# A Quality Integrated Fuzzy Inference System for the Reliability Estimating of Fluorochemical Engineering Processes

**Feng Xue [1], Xintong Li [1], Kun Zhou [1], Xiaoxia Ge [2], Weiping Deng [2], Xu Chen [1] and Kai Song [1,*]**

1   School of Chemical Engineering and Technology, Tianjin University, Tianjin 300350, China;
    3014207238@tju.edu.cn (F.X.); 3014207218@tju.edu.cn (X.L.); kzhou@tju.edu.cn (K.Z.); xchen@tju.edu.cn (X.C.)

2   Health, Safety and Environmental Protection Department, Juhua Group Co., Ltd., Quzhou 324004, China;
    ahbgxx@juhua.com.cn (X.G.); jhdwp@juhua.com.cn (W.D.)

*   Correspondence: ksong@tju.edu.cn; Tel.: +86-189-2031-7821

**Abstract:** Hypertoxic materials make it critical to ensure the safety of the fluorochemical engineering processes. This mainly depends on the over maintenance or the manual operations due to the lack of precise models and mechanism knowledge. To quantify the deviations of the operating variables and the product quality from their target values at the same time and to overcome the measurement delay of the product quality, a novel quality integrated fuzzy inference system (QFIS) was proposed to estimate the reliability of the operation status as well as the product quality to enhance the performance of the safety monitoring system. To this end, a novel quality-weighted multivariate inverted normal loss function was proposed to quantify the deviation of the product quality from the target value to overcome the measurement delay. Vital safety process variables were identified according to the expert knowledge. Afterward, the quality loss and the vital variables were inputs to an elaborate fuzzy inference system to estimate the process reliability of the fluorochemical engineering processes. By integrating the abundant expert knowledge and a data-driven quality prediction model to design the fuzzy rules of QFIS, not only the operation reliability but also the product quality can be monitored on-line. Its superiority in estimating system reliability has been strongly proved by the application of a real fluorochemical engineering process located in East China. Moreover, the application of the Tennessee Eastman process also confirmed its generalization performance for other complicated black-box chemical processes.

**Keywords:** process reliability estimating; fluorochemical engineering process; fuzzy inference system; quality prediction; prognostics and health management

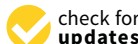



## 1. Introduction

With the diversification of products and the continuous development of application fields, the fluorochemical industry has become more and more important. However, hypertoxic materials widely exist in the fluorochemical engineering process. Even a tiny leak of these hypertoxic materials in the environment would cause huge damage to people, equipment and even public safety. Additionally, nowadays, the operation condition of chemical industrial processes is typically monitored by a large number of different types of sensors, capturing temperature, pressure, flow, vibration, solution concentration and other process variables. This not only results in very heterogeneous data at different time scales but also introduces the signals affected by measurement and transmission noise. In many cases, consequently, the sensors are partly redundant or highly related variables. Failures in such redundant sensors would not cause the same influence on the operation reliability as what a vital process variable would cause. Therefore, the requirement of the monitoring system goes far beyond fault detection and diagnosis, whose major tasks are limited to react after there are failures or faults happen. In order to avoid any possible failures or faults, and to reduce maintenance costs and equipment uptime at the highest level, proactive maintenance measures should be taken. This means that the maintenance

strategy should swift from fault detection and diagnosis (FDD) to Prognostics and Health Management (PHM).

The goal of PHM is to provide methods and tools to design optimal maintenance policies for a specific process under its distinct operating and degradation conditions, achieving a high availability at minimal costs [1,2]. It is not limited to the predictions of failure times or the remaining useful life (RUL) and supports optimal maintenance and logistics decisions by considering the process operation status, the operating context and the economic consequences of different faults. System reliability estimating (SRE) plays an important role in PHM. It focuses on assessing the operational reliability based on outputs from process operation status, available resources and operational demand.

SRE is developing rapidly and there are many methods currently available [3]. Fault tree analysis (FTA), reliability graph, Monte Carlo Simulation and Bayesian Networks (BN) are commonly used methods for it [4]. FTA is a systematic way to obtain the reliability of complex systems both qualitatively and quantitatively by using exact values of root causes' occurrence probability. Fuzzy methods are often applied in FTA to make up for the shortcomings of insufficient probability values [5,6]. Monte Carlo Simulation-based tools are useful for reliability assessment of large and complex power systems [7], but they may lead to a combinatorial explosion of the number of states to model a system [8]. They are more suitable for estimating the reliability of a component or system of low complexity rather than of highly complex systems [9].

Due to the complicated mechanism, copyright protection, hypertoxicity and so on, the lack of mathematical models is one of the major reasons hindering the applications of advanced control and monitoring methods in fluorochemical engineering processes. On the other hand, plenty of background and expert knowledge has accumulated along with the continuous operation of such processes. When there are differences in expert knowledge, expert consistency prioritization can be conducted for expertise differences instead of assuming experts identical or assigning some predefined weights [10]. Currently, because of the aforementioned reasons, over maintenance strategies and manual operations are the most commonly used strategies in the safety management of the fluorochemical engineering processes [11]. It is obvious that these methods lead to big economic losses and security risks [12]. Therefore, it is very urgent to propose an appropriate SRE method for further application of the PHM system in these processes.

Unlike principal component analysis (PCA) and other multivariate statistical process monitoring (MSPM) methods [13,14], a fuzzy inference system (FIS) can integrate data-driven modelling and the priceless expert knowledge by the designation of membership functions and fuzzy rules [15]. Successful applications of it are also attributable to its superiority to manage uncertainty and computation for noisy and imprecise data. It also takes advantage of operational experience and provides suggestions on chemical processes without hard intervention [16,17]. The success of FIS is evident from its applicability and relevance in extensive research areas: control systems, engineering, medicine, chemistry, finance and business, computer networks, computational biology, fault detection and diagnosis and pattern recognition [18,19]. It holds high promise in the realization of SRE for complicated and black-box processes like the fluorochemical engineering processes.

Additionally, an important aspect of any industrial operation is conformance to standards. This relates to how closely the operational performance, process safety, as well as quality of the final products, match the design specifications. Whether the product quality matches the expected value is an important standard to estimate how healthy the process operation status is. However, quality control has not been taken into consideration when the SRE or PHM system was designed. It is very important for decision-makers to know the overall status both reflects the safety assessment and the product quality of the chemical process to make the best response. Unfortunately, for chemical engineering industries, there is always an intolerably long time-delay in the measurement of product quality. This paper, therefore, is aimed to propose a quality integrated SRE method to fill in the gap between the PHM system and quality control.

Therefore, a quality integrated FIS model (QFIS) based system reliability estimating method was proposed and applied to a process in a fluorochemical factory located in East China. A novel quality loss function was proposed to estimate the quality deviation based on a product quality regression model trained with the Partial Least Squares (PLS) algorithm to overcome the time-delay in quality measurement. Meanwhile, vital safety variables of the fluorochemical process operation were selected under the guidance of expert knowledge. Then, these vital safety variables and the quality loss value were used as inputs to the FIS model. By making good use of the expert knowledge and the operation experience, the membership functions and fuzzy rules were well-constructed to obtain the system reliability of the fluorochemical engineering process. To test the generalization ability of our proposed QFIS method, it was also used in the Tennessee Eastman process, a widely applied benchmark for advanced control and monitoring system.

The rest of the paper is organized as follows: Section 2, brief introduction of R22 refrigerant producing process and existing algorithms used in this paper; Section 3, details of the proposed quality integrated fuzzy inference system; Section 4, applications in the R22 refrigerant producing process and the Tennessee Eastman process; Section 5, conclusion.

## 2. Background and Methods

### 2.1. Brief Introduction of R22 Refrigerant Producing Process

R22, also known as HCFC-22, is one of the most widely used fluorides. It is mainly used as a kind of common propellant and refrigerant. The global use of R22 continues to increase because it is a versatile intermediate in the organic fluorine chemical industry, e.g., as a precursor to tetrafluoroethylene.

The producing process of R22 is presented in Figure 1. The main operating units include a Feeder, a Reactor, a Water Scrubber, a Separator and two Rectifying columns. R22 is prepared from the chloroform as: $HCCl_3 + 2\,HF \rightarrow HCF_2Cl + 2\,HCl$.

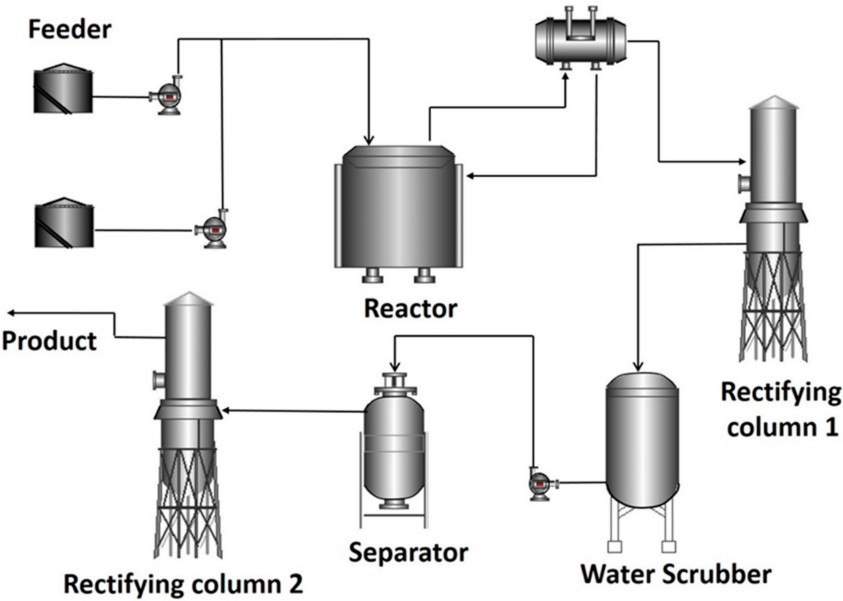

**Figure 1.** The producing process of R22.

All materials and byproducts like AHF, HCL and HF become intensely corrosive when meeting water in the air. Therefore, it is very vital to public safety and environmental protection to secure the safety of the R22 producing process and to improve the performance of the PHM system of it. As mentioned above, however, the complicated characteristics, the confidential agreement, and the time-varying mechanisms of it adversely hinder the performance of the traditional FDD methods. Over maintenance and manual operations are still the most commonly used strategies in the safety management of it.

The most important thing is that even a tiny amount of material or byproduct leakage into the environment can cause terrible damage to equipment and workers. PHM or at least SRE should play a bigger role than the traditional FDD, which can only react after a disaster happens, in predicting the operation stability to react and prevent a disaster from happening.

On the other hand, the sampling interval of the product quality is 180 min while the DCS process variables are 1 min. Such a long delay in product quality measurement is completely intolerable for quality control and safety management practice. It is not practical to use product quality as an input to estimate its deviation neither.

Therefore, we proposed a quality integrated fuzzy inference system to integrate the quality control and the system stability estimating at the same time for further applications of PHM in such a complicated and hypertoxic process. It also can overcome the time delay in quality measurement to evaluate the process operation through the quality control point of view.

### 2.2. Brief Introduction of Existing Algorithms

#### 2.2.1. Partial Least Squares (PLS)

Partial Least Squares (PLS) is a widely used linear regression method. It aims at modeling linear relationships between the input variables $X \in \mathbf{R}_{m \times n}$ ($n$ is the number of process variables and $m$ is the number of observations) and output variables $Y \in \mathbf{R}_{m \times p}$ ($p$ is the number of output variables) [20]. Regularly, $X$ and $Y$ are supposed to be normalized. The PLS model structure can be described as [21]:

$$Y = XB + V \tag{1}$$

where $B \in \mathbf{R}_{n \times p}$ is the regression coefficient matrix and $V \in \mathbf{R}_{m \times p}$ is the residual matrix.

It iteratively extracts the Latent Variables (LVs) $t_i \in \mathbf{R}_m$, $u_i \in \mathbf{R}_m$ and the weight vectors $w_i \in \mathbf{R}_n$, $c_i \in \mathbf{R}_p$ from $X$ and $Y$ matrices in decreasing order of their corresponding singular values, where $i = 1, \ldots, v$, and $v$ is the number of LVs, which is usually determined by cross-validation. In other words, the PLS algorithm decomposes $X$ and $Y$ matrices as follows [22]:

$$X^{\mathrm{T}} = \sum_{i=1}^{v} t_i p_i^{\mathrm{T}} + E = TP^{\mathrm{T}} + E \tag{2}$$

$$Y^{\mathrm{T}} = \sum_{i=1}^{v} u_i q_i^{\mathrm{T}} + F = UQ^{\mathrm{T}} + F \tag{3}$$

where $E$ and $F$ are the residual matrix of $X$ and $Y$, respectively. Therefore, by extracting LVs, the $n$-dimensional original input space $X$ is compressed into the $v$-dimensional LV-space. In common cases, $v << n$. By doing this, PLS can effectively remove the noise and multi-collinearity of the original data, which is especially true for the chemical process data [23].

Then the estimated regression coefficient matrix can be obtained by the following Equation:

$$\hat{B} = X^{\mathrm{T}} U \left( T^{\mathrm{T}} X X^{\mathrm{T}} U \right)^{-1} T^{\mathrm{T}} Y \tag{4}$$

The absolute value of the coefficient represents the contribution of the corresponding variable to the linear model, therefore, it can be used to quantify the importance of the corresponding variable.

Root mean square error (RMSE) is usually used as a metric on the determination of the value of $v$ in the PLS model. The definition of RMSE is:

$$\mathrm{RMSE} = \sqrt{\sum (\hat{y}_i - y_i)^2 / n} \tag{5}$$

where $y_i$ is the observed value of the output variable and $\hat{y}_i$ is the corresponding predicted value. PLS model performed better when the RMSE of the training data and testing data were smaller.

### 2.2.2. Fuzzy Inference System (FIS)

A fuzzy inference system (FIS) is a tool for modeling a complex system without a thorough mathematical explanation [24]. It is capable of modeling uncertainties commonly represented in linguistic form and extending the functionality of the engineering system. The term "fuzzy" refers to the fact that the involved logic can deal with concepts that cannot be expressed as "true" or "false" but rather as "partially true" or "partially false". The design of fuzzy rules is a delicate task, and it can be generally carried out by an expert, who, on the basis of some heuristics that s/he has developed about the system. This makes it easier to mechanize tasks that are already successfully performed by humans.

The application procedure of a FIS system consists of three steps: an input stage, a processing stage, and an output stage [15,25]:

(1)     The input stage maps inputs to the appropriate membership functions and true values. The most common shape of membership functions is triangular, although trapezoidal and bell curves are also used. The shape is generally less important than the number of curves and their placements.
(2)     The processing stage invokes each appropriate rule and generates a result, then combines the results of all rules. It is based on a collection of logic rules in the form of IF-THEN statements, where the IF part is called the "antecedent" and the THEN part is called the "consequent". Typical a fuzzy control system has dozens of rules.
(3)     The output stage converts the combined result into a specific control output value.

## 3. The Quality Integrated FIS Based System

PHM aims to provide an integrated framework for degradation prediction and system maintenance [2,26]. Since PHM can be considered as a holistic approach to an effective and efficient system health management, the quality of the produced products of a concerned process should be taken into serious consideration. However, most proposed methods and tools mainly focus on operation stability and maintenance cost. On the other hand, as aforementioned, for fluorochemical engineering and other chemical industries, there is always a long time-delay in the measurement of product quality. For example, in the R22 refrigerant producing process, the sampling interval of the product quality is 180 min while the DCS process variable is 1 min. It is not practical, in other words, to use product quality as an input to estimate its drift. Therefore, a novel quality integrated FIS (QFIS) based system reliability estimating method was proposed to evaluate the stability of the operation status and the quality of the product simultaneously.

### 3.1. A Novel Quality Weighted Multivariate Inverted Normal Loss Function (QMINLF)

Inverted normal loss function (INLF), Modified inverted normal loss function (MINLF), Inverted Beta loss function (IBLF) and Inverted Gamma loss function (IGLF) are different loss functions considering a random deviation from target values and are widely used in industrial applications [27].

For a multivariate process, to consider the deviation of a variable from its expected value as well as the importance of it to the final product, a novel quality weighted multivariate inverted normal loss function (QMINLF) was proposed as:

$$L(Q) = \frac{1}{1 - e^{-\rho^2}} \sum_{i=1}^{n} \left( 1 - e^{-\frac{1}{2}\beta_i(x_i - a_i)^2} \right) \tag{6}$$

where $L(Q)$ is the estimated quality loss, $\rho$ is a shape parameter, $n$ is the number of process variables, $x_i$ ($i = 1, 2, \ldots, n$) is the observed value of the $i$th process variable, $a_i$ is its

expected value and $\beta_i$ is the corresponding importance index. $\beta_i = \frac{|b_i|}{\sum_{i=1}^{n}|b_i|}$ and $|b_i|$ is the absolute value of the $i$th element value of $\hat{B}$ in Equation (4).

From the definition of $L(Q)$, we can see that $L(Q)$ is decided by the deviation values $(x_i - a_i)^2$ of all quality-related process variables. The bigger the deviation, the bigger the $L(Q)$ value. Additionally, for the same deviation value, because of being weighted by $\beta_i$, the more important the process variable $x_i$ to the final quality, the bigger the $L(Q)$ value, which means the bigger the quality loss. Therefore, QMINLF is more sensitive to the deviations of the comparatively higher quality-related process variables.

Note: $\rho$ should be optimized according to the operation knowledge or the performance to make sure $0 \leq L(Q) \leq 1$. It also can be optimized by a genetic algorithm or other optimization methods.

### 3.2. The Procedure of the Proposed QFIS Method

To make good use of operation knowledge and to consider the quality loss at the same time, a quality integrated fuzzy inference system was proposed to estimate the system's reliability. The procedure of it consists of four major steps:

(1)　Identifying the vital safety variables to process stability based on operation experience and background knowledge;
(2)　Quantifying the importance of variables to product quality using PLS algorithm;
(3)　Estimating the quality loss according to Equation (6);
(4)　Designing the membership functions and fuzzy rules for operational reliability using quality loss and vital safety variables as inputs.

The designing of the membership functions is the most time-consuming step for the QFIS method, and it is also the most important step to make sure the performance of the QFIS method. It is supposed to integrate the operation experience, background knowledge and mechanism analysis in this step. Therefore, the membership functions should be designed specifically. Strict membership functions are more preferred when the system is designed to provide warning of system reliability, so triangular and trapezoid are suggested as the membership curves. The details of membership functions and fuzzy rules construction will be given with the specific cases in Section 4.

The structure of the proposed method is presented in Figure 2 and the details are listed in Table 1.

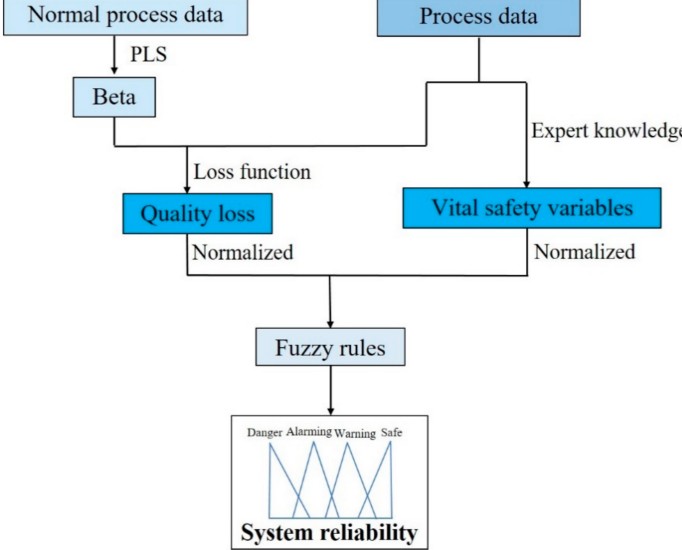

**Figure 2.** The structure of the quality integrated FIS.

**Table 1.** The procedure of the quality integrated fuzzy inference system (FIS).

| The Procedure of the Quality Integrated FIS |
| --- |
| **Vital safety variables identification**<br>    Step 1: Determining important operating unit and their process variables;<br>    Step 2: Selecting vital safety variables under the guidance of expert knowledge; |
| **Quality loss estimation (after collecting a certain number of normal observations)**<br>    Step 1: Quantifying the importance of quality-related process variables to product quality using PLS;<br>    Step 2: Estimating quality loss using loss function in Equation (6); |
| **Fuzzy inference system**<br>    Step 1: Normalizing the quality loss and vital safety variables as the inputs;<br>    Step 2: Constructing fuzzy rules and membership functions;<br>    Step 3: Obtaining the system reliability of the chemical process; |

## 4. Application Results and Discussion

### 4.1. Application in R22 Refrigerant Producing Process

To test the performance of the proposed QFIS method, the observations of a part of an R22 producing process in a large-scale fluorochemical industry company located in East China were applied. The flowchart of the R22 producing process is shown in Figure 1. There are 69 process variables in total. The sampling period was from May to November 2019 (Sampling interval was one minute), which included dozens of procurement cycles of raw materials and experienced through the summer, autumn and winter of the location.

According to the operation experience, as shown in Figure 3, data in the green box were observations in normal operation status which were used as training data. Data in red boxes were observations corresponding to three types of abnormal status. Ten thousand observations for each of them were used to test the performance of the proposed method.

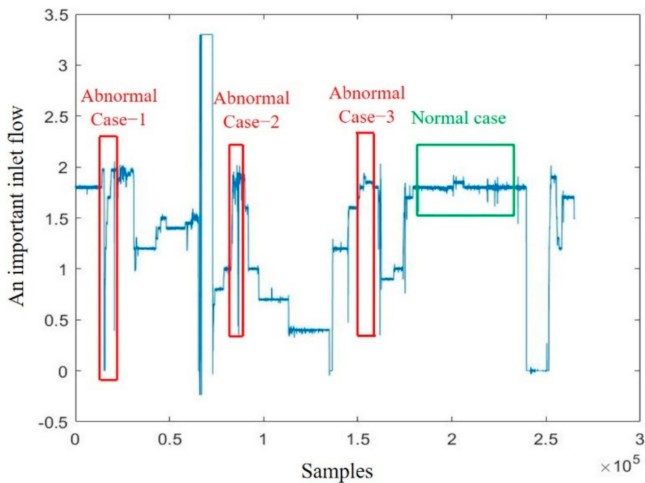

**Figure 3.** The selection of training and testing data for R22.

The purity of one of the major intermediate materials was used as the quality variable. Because it is sampled and measured offline per hour while the DCS process variables are 1 min, therefore, it is not practical to use product quality as an input to estimate its drift. To match with the measured product quality, only the process variables sampled at the same time were used to train the QFIS model. In this way, 849 observations of normal operation status were available. Of the observations, 699 were used as training data while the other 150 observations were used as testing data of the PLS-based quality prediction model. Three-fold cross-validation was performed in the training step to optimize the quality prediction model so that the importance index of each process variable can be evaluated comprehensively.

Besides the reactor, which is always the major unit of a process, rectifying column 1 is another major one suffering from corrosion according to the operation records and the expert knowledge. Consequently, the reactor level, reactor temperature, rectifying column 1 pressure and rectifying column 1 level were selected as vital safety variables. Then these four vital safety variables and quality loss value calculated by Equation (6) were normalized and input to the QFIS model to estimate the operation reliability.

As we mentioned above, membership functions or fuzzy rules are very important. The membership functions were determined strictly according to the characteristics of each input, and was basically divided into three levels: high, medium and low reliability. According to the expert knowledge and experimental experience, triangle and trapezoid curves were used. The membership functions for each input and output is presented in Figure 4a. For the reactor temperature, higher than the up limitation would cause much worse damage than lower than the low limit. Then the level of reactor temperature was defined as Not-high and High. For the reactor level, Rectifying column 1 pressure and Rectifying column 1 level, low pressure or level has the potential risk of leakage and high pressure or level would bring damage to the equipment. Then the levels of them were defined as Low, Medium and High. For the quality loss, obviously, normally, it is supposed to be as low as possible. Medium or high-quality loss will bring huge economic loss which is unacceptable. So the level of quality loss was defined as Low and Not-low.

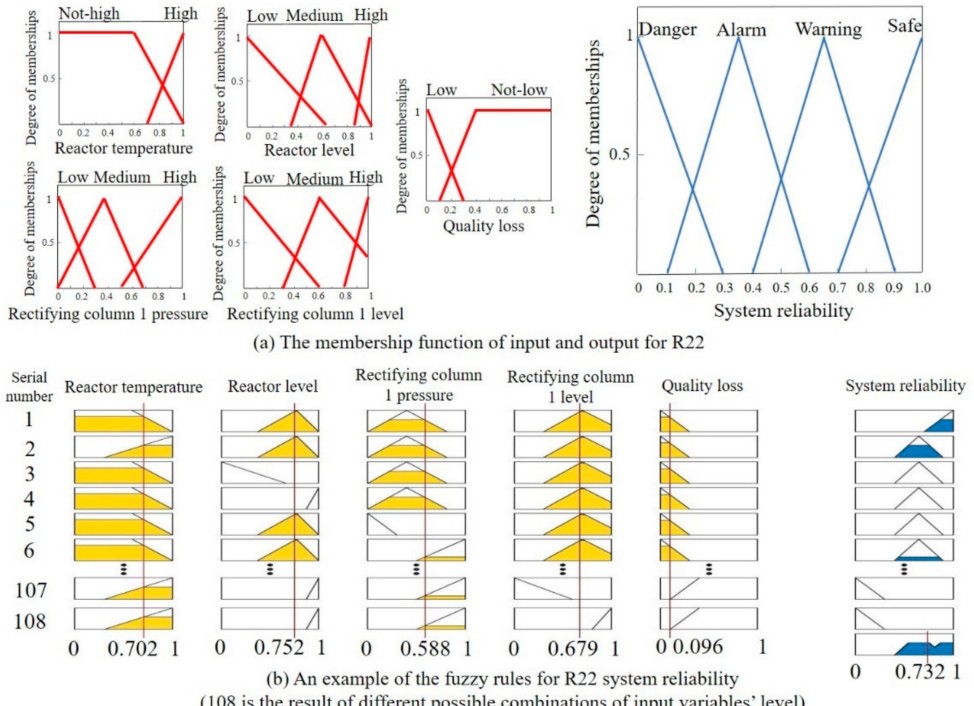

**Figure 4.** The examples of the application of the proposed method in R22 producing process.

The shape and position of membership function curves were determined according to the range of the corresponding variables and the operation experience which plays an important role due to the particularity of the R22 producing process. For example, according to equipment condition and operating status in the field. The reactor would experience a strong decrease in reliability when the reactor temperature is over 0.7. Then the starting point of the high reactor temperature curve is set as 0.7. The vertex of the medium reactor level curve (0.63) was the average value of reactor level in normal observations.

The fuzzy rules were determined according to the level of each input. Table 2 lists the level of each input and whether it is allowed. They were formulated as follows:

(1) If there was no input in state "No", then the system reliability was in "safe" status;

(2) If one or two inputs were in state "No", then the system reliability was in "warning" status;

(3) If three or four inputs were in state "No", then the system reliability was in "alarm" status;

(4) If there were five inputs in state "No", then the system reliability was in "danger" status.

**Table 2.** Constraints of inputs of R22 refrigerant producing process.

| Index | Low | Medium | High |
|---|---|---|---|
| Reactor temperature | | Yes | No |
| Reactor level | No | Yes | No |
| Rectifying column 1 pressure | No | Yes | No |
| Rectifying column 1 level | No | Yes | No |
| Quality loss | Yes | | No |

For example, if Reactor temperature was not high, Reactor level was medium, Rectifying column 1 pressure was high, Rectifying column 1 level was low and Quality loss was low, then the system reliability was in "warning" status.

A total of 108 fuzzy rules were constructed. Parts of them are presented in Figure 4b.

Then, the estimated system reliability of three abnormal cases is shown in Figure 5 with the corresponding value of an important inlet flow (IIF), which partly indicated the operation status of the process.

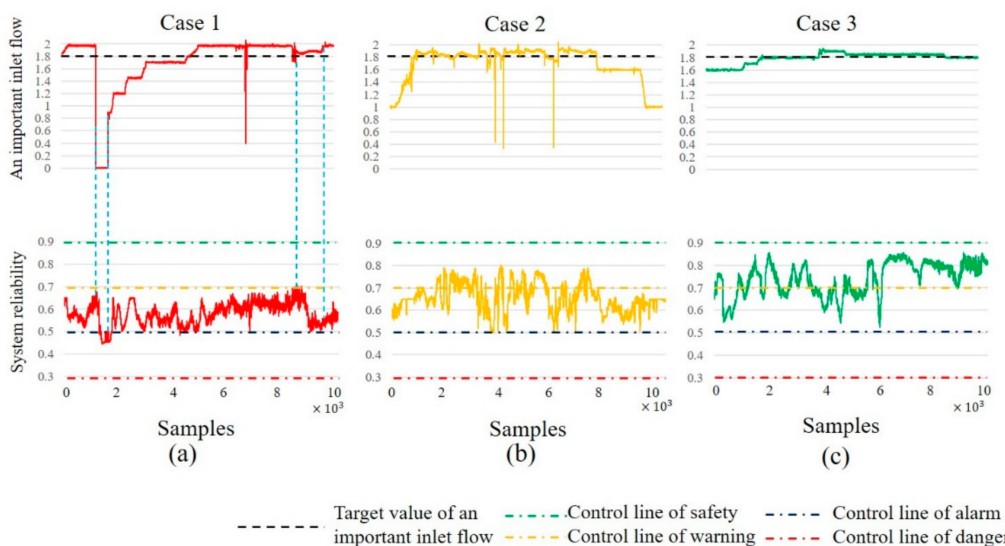

**Figure 5.** The important inlet flow and system reliability of three abnormal cases in R22; (**a**–**c**) are three abmormal cases' important inlet flow and system reliability.

In fact, the valve of the important inlet had to be shut down for a while to protect the process. Correspondingly, there was a very big deviation in the IIF value of Case 1 from its target value (1.8). As a result, there was a big fluctuation in the system reliability curve with a delay of only 40 min (see the period between the first two blue lines in Figure 5). The system reliability was continuously lower than 50% triggered a "Danger" notice. During the recovering period of the process, the overall trend of the system reliability was upward volatility because the spread of this shutdown influenced other vital variables. The system reliability was recovered to "Alarm", then "Warning". Unfortunately, another big disturbance happened at the end of the period of Case 1, and the reliability went back to "Alarm" status again. The system reliability was always under the control line of warning

which represented that the system was not reliable and the "shut down" of the important inlet was necessary.

Case 2 was during the recovering period of the process after an overhaul. The system reliability was going upward at the beginning. The ups and downs of the reliability had the same trend as the IIF value, and both of them were because of the operation fluctuations caused by the DCS (distributed control system). It was obvious that the system reliability was around the control line of warning and was improved in the recovering period.

Case 3, the operation was almost recovered back to normal with only small fluctuations. The system reliability was going upward toward 90% with fluctuations too. Most of the observations were above the control line of warning. It showed that the system was reliable.

With the lowest average reliability of Case 1 and the better system reliability in Case 2 and Case 3, the same trends between the system reliability and the IIF value strongly proved the performance of our method. It can predict the system reliability with a reasonable delay using only normal observations to train the model. The proposed method comprehensively considered both safety and economic factors, and the result fully reflected the status of the system, so as to provide appropriate suggestions to the decision-makers of the R22 refrigerant producing process operation.

Due to the confidential agreement, it is not allowed to show the quality variables of the R22 producing process. The contribution of quality loss to the final estimated system reliability would be discussed in the application in the Tennessee Eastman process in Section 4.2.

### 4.2. Application in the Tennessee Eastman Process

To further test the proposed method, it was applied to predict the system reliability of the Tennessee Eastman chemical process (TEP). TEP is a chemical simulation process that was promoted by J. J. DOWNS and E. F. VOGEL in 1992. It consists of five major operating units namely, a reactor, a product-condenser, a vapor-liquid separator, a recycle compressor and a product stripper. Its process flow diagram is shown in Figure 6. G and H are products of the TEP. Twenty-two process variables and twelve manipulated variables are measured online. Among them, two manipulated variables are constant. Additionally, nineteen variables are measured by offline equipment. There are 28 process fault types (IDV1-IDV28) in the revised version for researchers to test their monitoring methods [28,29]. Details of them are available in Refs. [28,29].

According to the literature research, the reactor pressure, reactor level, product separator level and stripper base level were selected as the vital safety variables. Normally, the reactor temperature should be considered as a vital variable, but it will not be affected by any available IDVs provided by TEP. Therefore, it was not selected as a vital variable.

The 22 process variables and 10 non-constant processes manipulated variables were used as the input of the PLS-based quality prediction model. The ratio of the product G and H was used as the output of this quality model. A total of 960 observations, which were sampled per 3 min, were used as training data and 240 observations were used as testing data. The longest time delay in the G/H ratio was 30 min, which was ten times of the process variables. Five-fold cross-validation was taken to optimize this model. The result showed when $v = 2$, both the RMSEs of training data and testing data had the smallest values. It meant the quality model had the best regression performance when $v = 2$. So the corresponding regression coefficient vector $\beta$ was used to obtain the quality loss defined in Equation (6).

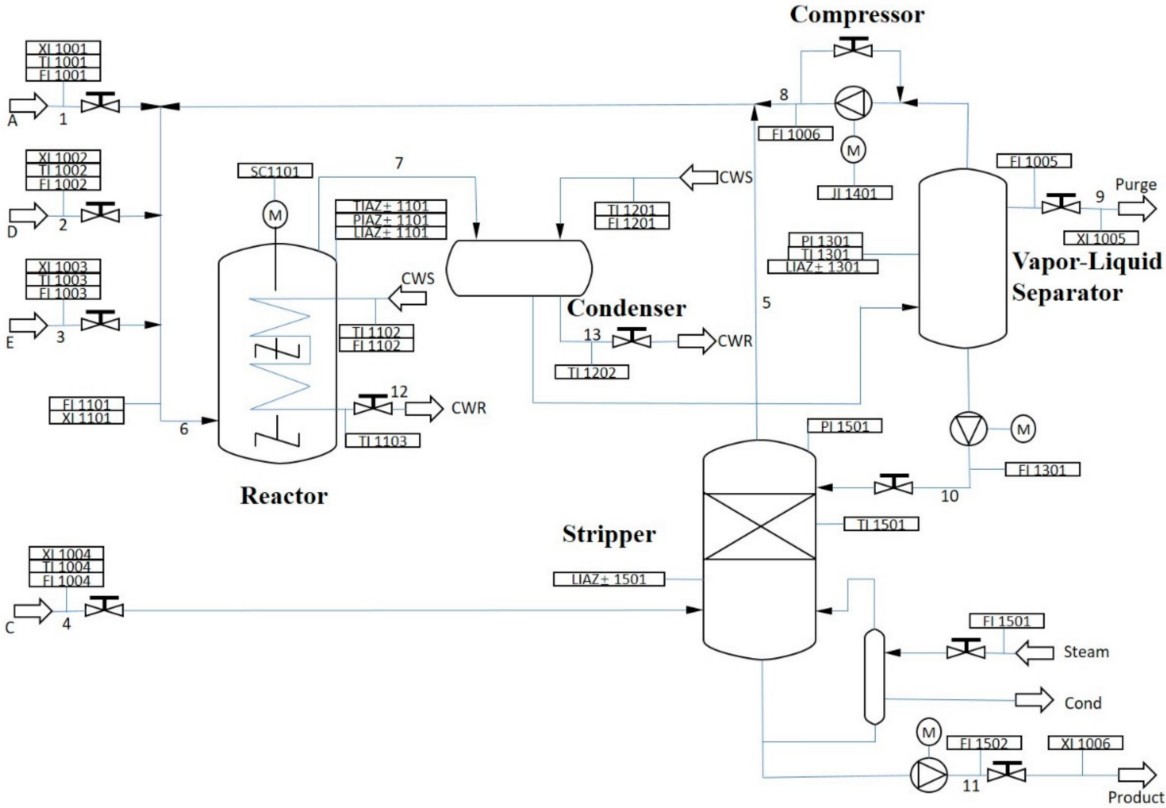

**Figure 6.** The producing process of Tennessee Eastman process.

The principles of designing membership functions and fuzzy rules were similar to those of the R22 refrigerant producing process. The constraints of the five inputs of QFIS are presented in Table 3. A total of 72 fuzzy rules were designed. The membership functions of input and output are presented in Figure 7a and part of the fuzzy rules viewer is presented in Figure 7b. Details of the codes for TEP is provided in the supplementary material at the end of the paper.

**Table 3.** Constraints of inputs of Tennessee Eastman process.

| Index | Low | Medium | High |
|---|---|---|---|
| Reactor pressure | | Yes | No |
| Reactor level | No | Yes | No |
| Product separator level | | Yes | No |
| Stripper base level | No | Yes | No |
| Quality loss | Yes | No | |

These 28 IDVs occur in a different part of the TEP with different characteristics and amplitudes, they cause different influences on the process operation. According to the system reliabilities calculated with our method, these 28 IDVs can be divided into three categories: (1) Low danger: The system reliabilities were minor impacted by some IDVs and they were always around 90%; (2) Medium danger: Some IDVs had a middle impact on the safe operation of TEP and the corresponding system reliabilities were between 60% and 90%; (3) High danger: A few IDVs impacted the operation severely and the system reliabilities were under 50%. The IDV descriptions of TEP and their danger levels were given in Table 4. We selected three typical IDVs as examples to demonstrate our QFIS method. The system reliabilities and corresponding ratio of G/H and quality loss are shown in Figure 8.

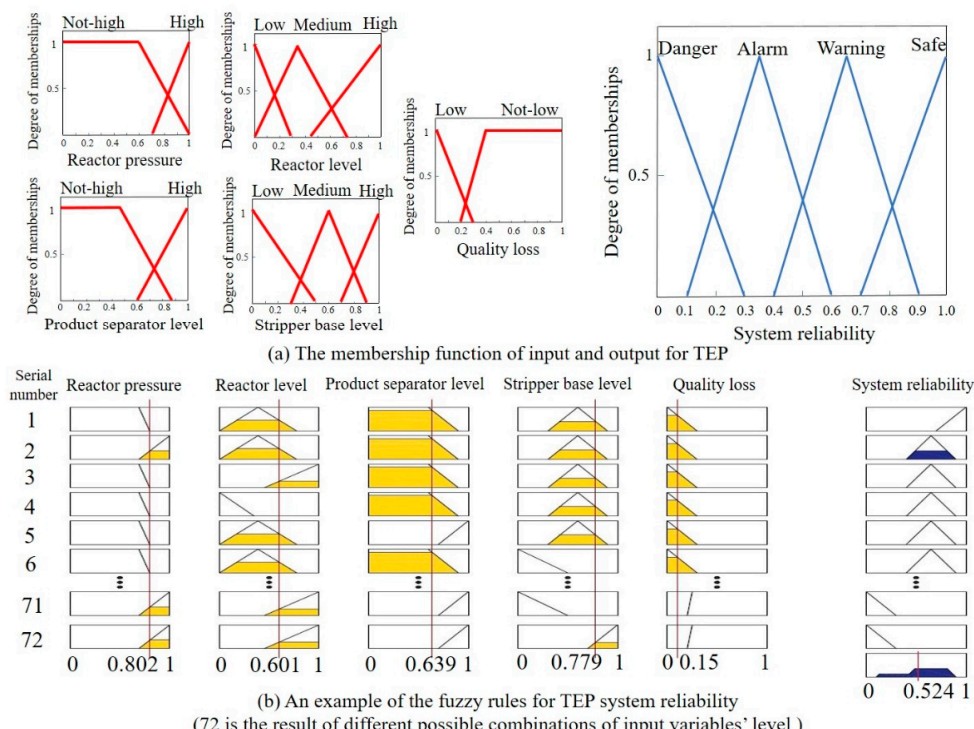

**Figure 7.** The example of the application of the proposed method in the Tennessee Eastman chemical process (TEP).

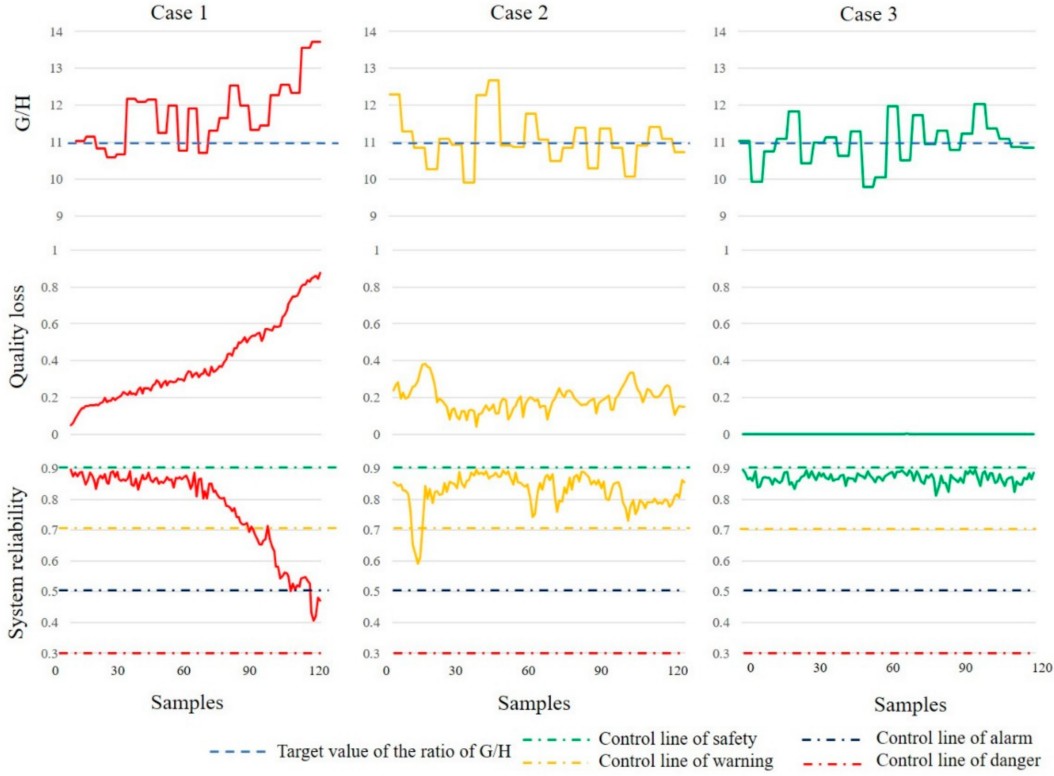

**Figure 8.** The ratio of G/H, quality loss and system reliability of three cases in TEP.

**Table 4.** The fault description of Tennessee Eastman Process.

| No. | Description | Type | Danger Level |
| --- | --- | --- | --- |
| 1 | A/C feed ratio, B composition constant (stream 4) | Step | Medium |
| 2 | B composition. A/C ratio constant (stream 4) | Step | Low |
| 3 | D feed temperature (stream 2) | Step | Low |
| 4 | Reactor cooling water inlet temperature | Step | Low |
| 5 | Condenser cooling water inlet temperature | Step | Low |
| 6 | A feed loss (steam 1) | Step | High |
| 7 | C header pressure loss-reduced availability (steam 4) | Step | Low |
| 8 | A, B, C feed composition (stream 4) | Random | Medium |
| 9 | D feed temperature (stream 2) | Random | Low |
| 10 | C feed temperature (stream 4) | Random | Low |
| 11 | Reactor cooling water inlet temperature | Random | Low |
| 12 | Condenser cooling water inlet temperature | Random | Medium |
| 13 | Reaction kinetics | Slow drift | Medium |
| 14 | Reactor cooling water valve | Sticking | Low |
| 15 | Condenser cooling water valve | Sticking | Low |
| 16 | Unknown | Unknown | Low |
| 17 | Unknown | Unknown | Low |
| 18 | Unknown | Unknown | Medium |
| 19 | Unknown | Unknown | Medium |
| 20 | Unknown | Unknown | Low |
| 21 | A feed temperature (stream 1) | Random | Low |
| 22 | E feed temperature (stream 3) | Random | Low |
| 23 | A feed pressure (stream 1) | Random | Low |
| 24 | D feed pressure (stream 2) | Random | Medium |
| 25 | E feed pressure (stream 3) | Random | Low |
| 26 | A and C feed pressure (stream 4) | Random | Low |
| 27 | Pressure fluctuation in the cooling water re-circulating unit of the reactor | Random | Low |
| 28 | Pressure fluctuation in the cooling water re-circulating unit of the condenser | Random | Low |

Case 1 was the IDV6 (A feed loss) in TEP. The loss of the main raw material of the reactor caused a severely bad influence on the product quality and the process operation. After IDV6 was introduced, the system reliability dropped very quickly to a lower 40%, which indicated that the process was in a very dangerous status. The quality loss increased quickly and the ratio of G/H deviated from its target value severely at the end of the simulation. Actually, the simulation would shut down in 6.2 h after IDV6 was introduced. This strongly proved how serious the damage was caused by IDV6. It also proved how good the performance of our system reliability estimating method.

Case 2 was the IDV12 (A random fluctuation in the condenser cooling water inlet temperature) in TEP. The condenser was not the major part of the TEP. Moreover, according to the mechanism and the flowchart of TEP, a random fluctuation in the condenser cooling water inlet temperature could not cause severe damage because of the time delay and the operation of the automatic DCS system. The product quality was only slightly affected and the simulation could still keep running. The ratio of G/H was in a reasonable fluctuation too. Consequently, the system reliability did not drop seriously. Except for several points that were lower than 70% (the control line of warning), it was around 90% most of the time.

Case 3 was IDV9 (A random fluctuation in the D feed temperature) in TEP. D was a reaction raw material in TEP. The automatic DCS system tuned the process parameters to overcome the influence caused by it. Therefore, it only caused mild fluctuations in the process operation. The quality loss was extremely small and the ratio of G/H was around the target value. The system reliability was around the control line of safety which meant the status was safe and reliable.

Therefore, the following conclusion can be summarized from the results: (1) The proposed method is sensitive to the change both in product quality and in the safety of TEP. (2) The degree of the abnormal status of TEP can be accurately estimated by our proposed method.

## 5. Conclusions

The proposed methodology considers both the effect of quality control and safety assessment on the reliability of the chemical process system. To estimate the quality deviation and to overcome the time delay in quality measurement, which is very common for engineering practice, a novel quality loss function was proposed by weighting the contributions of the process variables to the final product. Meanwhile, the vital safety variables of the fluorochemical process operation were selected under the guidance of expert knowledge. Finally, the system reliability was estimated with an elaborate fuzzy inference system using the quality loss and vital process variables as inputs. The membership functions and fuzzy rules were constructed by making good use of the expert knowledge and the operation experience. Applications on a practical fluorochemical engineering process in East China and on the Tennessee Eastman process strongly confirmed the superiority of QFIS in system reliability estimating of the proposed system for complicated black-box chemical processes. The most important contribution of the proposed methodology is to provide an overall system reliability assessment method on both quality control and operation status which can offer a comprehensive proposal on further PHM. However, the estimated system reliability which should serve as a reference for decision-makers can not control or regulate the chemical process directly. The result shows the overall status of the chemical process and more research needs to be carried out on identifying specific reasons leading to the decline in system reliability.

**Supplementary Materials:** The following are available online at https://www.mdpi.com/2227-9717/9/2/292/s1, MATLAB codes for TEP.

**Author Contributions:** Conceptualization, F.X. and K.S.; methodology, F.X. and K.S.; software, F.X.; validation, F.X., X.L. and X.G.; formal analysis, F.X. and X.L.; investigation, F.X.; resources, W.D.; data curation, F.X., X.L., X.G. and K.Z.; writing—original draft preparation, F.X.; writing—review and editing, K.S.; visualization, K.Z.; supervision, X.C.; project administration, K.S.; funding acquisition, K.S. All authors have read and agreed to the published version of the manuscript.

**Funding:** This work was supported by the National Key Research and Development Program of China (No. 2018YFC0808600), Ministry of Science and Technology of the People's Republic of China.

**Conflicts of Interest:** The authors declare no conflict of interest.

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
