# Peer review of "A Quality Integrated Fuzzy Inference System for the Reliability Estimating of Fluorochemical Engineering Processes"

_processes, doi:10.3390/pr9020292_

Round 1

Reviewer 1 Report

The abstract should not contain “meanwhile” (Line 18). “well-designed” is not a scientific phrase as there is no criteria provided (Line 20). What do you mean by “rich operating experience” (Line 21)? This is not a scientific language. “powerful enhancement” is also not appropriate.

The abstract is too long and needs to be rewritten to clearly present research problem, novelty, methods and contribution.

Line 178 contains grammar issues (extract)

Line 181: what is ‘mean zeros’?

Line 296 and others: samples should be changed into observations.

What is the difference between figures 4 and 7? They contain awkward terms: not-high, not-low, danger, alarm, warning (the latter three mean the same). The titles (“viewer”) are wrong. What is the meaning of 108?

Lines 315-325 should be turned into a text rather than a list. Obviously, the whole paper presents results of the research, so you don’t need to use sentence on line 315.

Lines 326-329 are unclear and should be completely rewritten. It is crucial to know how the fuzzy numbers were constructed. Such phrases as ‘operation experience’ do not tell anything.

Line 418: what is the meaning of ‘it showed‘?

What is the meaning of ‚unknown‘ in Table 4?

Lines 453-455 are meaningless.

Conclusions should discuss limitations of the paper.

The following paper on fuzzy analysis in maufacturing should be cited: Latysheva, O.; Rovenska, V.; Smirnova, I.; Nitsenko, V.; Balezentis, T.; Streimikiene, D. 2020 “Management of a sustainable development of machine-building enterprises: A sustainable development space approach” Journal of Enterprise Information Management. https://doi.org/10.1108/JEIM-12-2019-0419

Author Response

Question 1: The abstract should not contain “meanwhile” (Line 18). “well-designed” is not a scientific phrase as there is no criteria provided (Line 20). What do you mean by “rich operating experience” (Line 21)? This is not a scientific language. “powerful enhancement” is also not appropriate.

Response: We appreciate your comment. We have changed or deleted inappropriate words. The detailed revision can be found in lines 18-20, page 1.

Question 2: The abstract is too long and needs to be rewritten to clearly present research problem, novelty, methods and contribution.

Response: We appreciate your suggestion. We have rewritten the abstract to make the research problems, novelty, methods and contribution clearer and more concise. The detailed revision can be found in lines 10-26, page 1.

Question 3: Line 178 contains grammar issues (extract)

Response: Sorry for this reckless mistake, we have corrected it. The detailed revision can be found in lines 155, page 4.

Question 4: Line 181: what is ‘mean zeros’?

Response: Sorry for this confusing language. In regression problems and some machine learning algorithms, it is usually necessary to centralize and standardize the original data. Therefore, the mean values of them are supposed to be zero. We have modified this sentence. The detailed revision can be found in lines 152-153, page 4.

Question 5: Line 296 and others: samples should be changed into observations.

Response: Thank you for your comment. We have changed samples into observations.

Question 6: What is the difference between figures 4 and 7? They contain awkward terms: not-high, not-low, danger, alarm, warning (the latter three mean the same). The titles (“viewer”) are wrong. What is the meaning of 108?

Response: We appreciate your comment. Figures 4 and 7 show the detailed fuzzy rules and results of our method in two different applications: R22 producing process and TEP. Because these two processes are completely different to each other, the rules and the results are different to each other too.

The “Not-high”, “not-low”, “danger”, “alarm”, “warning are” are widely used fuzzy terms in fuzzy inference system to describe the fuzzy situations of inputs and outputs of the process. “Danger”, “alarm” and “warning” are widely used terms in fault detection and diagnosis (FDD) and PHM to describe abnormal status of the process. “Warning” indicates something is wrong but not very urgent or vital; “alarm” indicates some serious faults happened and it may be vital; “danger” means vital fault happened to the process, the emergency plan need to be initiated to avoid further economic or life lost.

108 is the result of different possible combinations of input variables’ levels. Reactor temperature and Quality loss have 2 levels and the other 3 input have 3 levels. Then . We have added this explanation in Figure 4.  

Question 7: Lines 315-325 should be turned into a text rather than a list. Obviously, the whole paper presents results of the research, so you don’t need to use sentence on line 315.

Response: Thank you for your comment. Lines 315-325 have been revised accordingly.

Question 8:Lines 326-329 are unclear and should be completely rewritten. It is crucial to know how the fuzzy numbers were constructed. Such phrases as ‘operation experience’ do not tell anything.

Response: Thank you for your precious suggestion. Due to the particularity of the chemical process, it is difficult to determine the threshold value of the variable through a fixed method. This is one of the reasons why we adopt the fuzzy method. The determination of the fuzzy numbers depends on operating experience. We have rewritten Lines 326-329 and explained it in the latest revision. The detailed revision can be found in lines 297-303, page 9.

Question 9:Line 418: what is the meaning of ‘it showed’?

Response: Sorry for this confusing statement. We have revised this sentence.

The detailed revision can be found in lines 390-392, page 11.

Question 10: What is the meaning of ‘unknown’ in Table 4?

Response: Tennessee Eastman process is a chemical simulation process. There are 28 types of process faults provided by the designer for FDD, system reliability estimation and other related research. To test the performance of the methods proposed by different users more strict, besides faults with specific characteristics (for example step or random), there are 5 faults whose characteristics are unknown to our users. Because of the copyright and the masked codes, they remain unknown to us.

Question 11: Lines 453-455 are meaningless.

Response: Thank you for your comment. We have deleted the meaningless sentence.

Question 12: Conclusions should discuss limitation of the paper.

Response: Thank you for your precious suggestion. We have discussed the limitation of the paper in the latest revision. The detailed revision can be found in lines 448-450, page 14.

Question 13: The following paper on fuzzy analysis in maufacturing should be cited: Latysheva, ; Rovenska, V.; Smirnova, I.; Nitsenko, V.; Balezentis, T.; Streimikiene, D. 2020 “Management of a sustainable development of machine-building enterprises: A sustainable development space approach” Journal of Enterprise Information Management. https://doi.org/10.1108/JEIM-12-2019-0419

Response: Thank you for your precious suggestion. The paper you mentioned has been cited in the revised version. The detailed revision can be found in lines 84, page 2.

Reviewer 2 Report

Manuscript Title: A Quality Integrated Fuzzy Inference System for the Reliability 2 Estimating of Fluorochemical Engineering Processes

Manuscript ID: processes-1089566

This study proposes a novel quality and safety integrated fuzzy inference system for system reliability estimation. It is applied for fluorochemical engineering processes. The general structure is shon in Figure 2.

The authors should revise the paper based on given major modifications.

  • The paper has some blocks of similar paragraphs. The similarity of the content is %25 if the bibliography is excluded. This is the biggest and the most important problem.
  • I would like to point out the first three paragraphs in the section 2.1. What are they?
  • Limitations and future directions should be given.
  • The tables should be designed well. Figures might be clear and readable.
  • More details about the experts and their profiles should be given.
  • More details should be given about the application of the proposed method. The authors should proivde the codings, parameters, and step-by-step application in the Appendix section. Therefore, researchers can re-apply your work to test the results, and apply your model for their own problems.
  • Fix the title of the section 3. (Systel). Please check the paper from the beginning to the end.

In general, the proposed approach is good and interesting. The paper requires a major revision.

Author Response

Question 1:The paper has some blocks of similar paragraphs. The similarity of the content is %25 if the bibliography is excluded. This is the biggest and the most important problem.

Response: Sorry for the reckless mistake. We have removed the redundant paragraphs and revised the manuscript very carefully to avoid the similarity. Thank you so much!

Question 2: I would like to point out the first three paragraphs in the section 2.1. What are they?

Response: These first three paragraphs were automatically added by the format editing system using by this journal. They weren’t written on purpose and we couldn’t do anything about it. But we are still sorry for the troubles they caused.

Question 3: Limitations and future directions should be given.

Response: Thank you for your precious suggestion. We have discussed the limitations and future directions in the latest revision. The detailed revision can be found in lines 448-452, page 14.

Question 4: The tables should be designed well. Figures might be clear and readable.

Response: Thank you for your precious suggestion. We have made the tables and figures clear and readable in the latest revision.

Question 5: More details about the experts and their profiles should be given.

Response: Thank you for your precious suggestion. We are very eager to provide more details about the experts and their profiles. However, due to the confidentiality agreement and the interests of the company, we can not disclose more details on experts and their profiles. Nevertheless, the method we proposed can be used as a reference for researchers to learn and improve by combining their own expert knowledge.

Question 6: More details should be given about the application of the proposed method. The authors should provide the codings, parameters, and step-by-step application in the Appendix section. Therefore, researchers can re-apply your work to test the results, and apply your model for their own problems.

Response: Because of the Confidentiality agreement, the data and details of R-22 process are not allowed to submit. We are sorry for it. But the codes using TEP is available on request.

Question 7:Fix the title of the section 3. (Systel). Please check the paper from the beginning to the end.

Response: Sorry for this reckless mistake, this problem has been resolved in the latest revision. We have checked the paper from beginning to the end to resolve similar reckless mistakes.

Round 2

Reviewer 1 Report

The paper can be accepted

Author Response

Question 1:The paper can be accepted

Response: We appreciate your comment. Your affirmation is a great encouragement to our work, we will persist in further research and strive for continuous progress.

Reviewer 2 Report

Manuscript ID: processes-1089566

Title: A quality integrated fuzzy inference system for the reliability estimating of fluorochemical engineering processes

Abstract: The hypertoxic materials make it critical to ensure the safety of the fluorochemical engineering processes. It is mainly depending on the over maintenance or the manual operations due to the lack of the precise model and mechanism knowledge. To quantify the deviations of the operation variables and the product quality from their target values at the same time and to overcome the measurement delay of the product quality, a novel quality integrated fuzzy inference system (QFIS) was proposed to estimate the reliability of the operation status as well as the product quality to enhance the performance of the safe monitoring system. To this aim, a novel quality weighted multivariate inverted normal loss function was proposed to quantify the deviation of the product quality from the target value to overcome the measurement delay of it. Vital safety process variables were identified according to the expert knowledge. Afterwards, the quality loss and the vital variables were inputs to an elaborate fuzzy inference system to estimate the process reliability of the fluorochemical engineering processes. By integrating the abundant expert knowledge and a data-driven quality prediction model to design the fuzzy rules of QFIS, not only the operation reliability but also the product quality can be monitored on-line. Its superiority in estimating system reliability has been strongly proved by the application on a real fluorochemical engineering process located in East China. Moreover, the application on Tennessee Eastman process also confirmed its generalization performance for other complicated black-box chemical processes.

Report

The authors partially revised the paper. After the following minor revisions, it is recommended to be accepted.

The authors can discuss and give the examples for the fault tree analysis (FTA) in the literature. For example, “Fault Tree Analysis of chemical cargo contamination by using fuzzy approach”, “A novel Real-Time Continuous Fuzzy Fault Tree Analysis (RC-FFTA) model for dynamic environment”, “Consistency control and expert consistency prioritization for FFTA by using extent analysis method of trapezoidal FAHP” and “Crankcase explosion for two-stroke marine diesel engine by using fault tree analysis method in fuzzy environment”

For the fifth concern of my previous review, you can use the similar example for the expert profiles: “An analysis for the effects of different approaches used to determine expertise coefficients on improved fuzzy analytical hierarchy process method”, “Shipping technology selection for dynamic capability based on improved Gaussian fuzzy AHP model” and/or “A root cause analysis for Arctic Marine accidents from 1993 to 2011”

For the sixth concern, please add the codes using TEP in the Appendix section.

In general, the paper is OK. After the above-mentioned minor revisions the paper can be accepted for publication.

Author Response

Reviewer 2

Question 1: The authors can discuss and give the examples for the fault tree analysis (FTA) in the literature. For example, “Fault Tree Analysis of chemical cargo contamination by using fuzzy approach”, “A novel Real-Time Continuous Fuzzy Fault Tree Analysis (RC-FFTA) model for dynamic environment”, “Consistency control and expert consistency prioritization for FFTA by using extent analysis method of trapezoidal FAHP” and “Crankcase explosion for two-stroke marine diesel engine by using fault tree analysis method in fuzzy environment”

Response: We appreciate your comment. We have discussed and given the examples for the fault tree analysis (FTA) in the latest revision. The detailed revision can be found in lines 59-62, page 2.

Question 2: For the fifth concern of my previous review, you can use the similar example for the expert profiles: “An analysis for the effects of different approaches used to determine expertise coefficients on improved fuzzy analytical hierarchy process method”, “Shipping technology selection for dynamic capability based on improved Gaussian fuzzy AHP model” and/or “A root cause analysis for Arctic Marine accidents from 1993 to 2011”

Response: Thank you for your precious suggestion. We have read the profiles you shared and learned a lot from them. In these profiles, expert consistency prioritization is conducted for expertise differences instead of assuming experts identical or assigning some predefined weights. It is a great method to reduce errors caused by different expert knowledge. However, expert knowledge in our paper was obtained from experienced field engineers who had consistent and reliable knowledge on the producing process. So it is not very necessary for us to apply similar method. Nevertheless, the methods in these profiles are great and we will learn from them in our further research. The detailed revision can be found in lines 71-73, page 2.

Question 3: For the sixth concern, please add the codes using TEP in the Appendix section.

Response: We appreciate your comment. It will occupy too much space if we add the codes using TEP in the Appendix section. So we sorted out the relevant codes of TEP and we will upload it as supporting material.
